# Optimism in Financial Markets: Stock Market Returns and Investor Sentiments

**Chiara Limongi Concetto [1,2] and Francesco Ravazzolo [1,3,]*** 

[1]    Faculty of Economics and Management, Free University of Bozen-Bolzano, 39100 Bolzano, Italy;
       Chiara.Limongi@economics.unibz.it
[2]    Sparkasse—Cassa di Risparmio, 39100 Bolzano, Italy
[3]    Centre for Applied Macro and Commodity Prices, BI Norwegian Business School, 0484 Oslo, Norway
[*]    Correspondence: francesco.ravazzolo@unibz.it

**Abstract:** This paper investigates how investor sentiment affects stock market returns and evaluates the predictability power of sentiment indices on U.S. and EU stock market returns. As regards the American example, evidence shows that investor sentiment indices have an economic and statistical predictability power on stock market returns. Concerning the European market instead, investigation provides weak results. Moreover, comparing the two markets, where investor sentiment of U.S. market tries to predict the European stock market returns, and vice versa, the analyses indicate a spillover effect from the U.S. to Europe.

**Keywords:** Bayesian econometrics; portfolio choice; sentiments; stock market predictability

## 1. Introduction

Optimism, also known as market sentiment, reveals the movements in the financial markets dictated by the psychological perception of determined operations or trades. This could create situations of mispricing, leading investors to lower returns than they expected. These movements in sentiment can conduct price distant from economic fundamentals and pose new research questions. For example, is optimism, and consequently pessimism, a factor of influence in financial markets? Accordingly, investor sentiment, which captures these fluctuations, is increasingly a topic of research relevance.

Several studies have been conducted in order to examine the presence and the effects of sentiment in financial markets. Before of an investment, investors behave differently. According to their propensity to the risk and the future expectations, they are divided into rational and irrational traders. Many individuals, defined irrational, in making decisions underreact or overreact to fundamentals and returns. Therefore, evaluation and decision-making are biased with the result of mispricing, i.e., moving from its fundamental value. Definitions as overconfidence, conservatism, and representativeness can explain the concept, but there is no academic consensus on a theory or a right formula to quantify it. We dedicate the next section to a literature review and discussion on what has been found on the relationship between investor sentiments and stock market returns.

The aim of this paper is to extend the research on investor sentiments and stock market returns in three directions. First, the majority of studies investigates this relationship with American stock markets, because of their financial significance and the higher likelihood to access the data. One of the few exception is (Fernandes et al. 2013) that provide an examination of the Portuguese market. This paper would like to contribute to the literature by analysing and comparing a strong and stable market like the U.S. to a smaller one, but with economic significance, like Europe.

Second, we apply Bayesian inference allowing us to set priors such as that the posterior distribution of the parameters of the predictive return regression can better learn from the data. This is particular useful when the sample size is small and priors help to reduce parameter uncertainty as in our European case.

Third, we evaluate the out-of-sample predictability power of investor sentiment acting on this association and interpret the economic effects of the findings. Using various indices, which measure sentiment both in an implicit and explicit way, the U.S. and the European market are studied over the periods 1990–2014 and 2001–2017, respectively. Extending previous evidence, we add sentiment indices as a further regressor to those typically considered in stock market predictability; see, for example, the list in (Welch and Goyal 2008). The forecasts in both examples start from the year 2008 because of its economical relevance due to the financial crisis. Further analyses compare the two markets to each other, searching for a spillover effect. In this case, investor sentiment of U.S. market tries to predict the European stock market returns, and vice versa.

As regards the American example, we find that sentiment indices have a negative impact on the stock market returns and provide accurate predictions of next month stock returns. Excluding it from the set of regressors decreases substantially the economic and statistical predictability power. With respect to the European market, evidence show weak findings. Only the Consumer Confidence Index provides in-sample evidence of predictability, but none gives out-of-sample more accurate predictions that the random walk in mean benchmark. Finally, the results show the presence of a spillover effect between the two markets. From an economic standpoint, Europe, which has been affected by globalisation and quick communication, is more prone to follow the influence of the American sentiment, because of the stronger U.S. economy.

The structure of the paper is as follows: Section 2 provides the literature review, deepening what is investor sentiment and diversifying between its measurements. Section 3 deals with the methodology, the empirical applications, and the relative results. Section 4 sums up the conclusion and suggests issues for future works.

## 2. Literature Review

This chapter provides a brief definition of investor sentiment, supported by theories, extended to behavioural reasons and effects; and a review of the empirical analyses on its relationship with stock markets.

### 2.1. Investor Sentiment

First of all, it is pivotal to define what investor sentiment is and why it has become more important in recent times. Investor sentiment is also known as *market sentiment* since it reveals the movements in the financial markets dictated by the psychological perception of determined operations or trades. Investors are subject to the sentiment of the market, i.e., to the belief about future expectations and investment risks that are not consistent with the statistical data or real facts. When the business performance is driven by emotions, a distortion of the price from its fundamental value occurs, entailing the risk in itself to be misunderstood from the investors and worsen the situation. Therefore, sentiment represents generally the attitude of economic agents, from consumers to investors, towards the market.

Barberis, Shleifer, and Vishny (Barberis et al. 1998) introduce an investor sentiment, focusing on overreaction and underreaction. They explain that information could be misleading. Indeed, optimistic announcements drive the investors to an exaggerated optimism about future news and, therefore, to overreaction, which leads stock prices to increase. Unfortunately, the following "news announcements are likely to contradict his optimism, leading to lower returns" (Barberis et al. 1998). This idea simply resumes the evidence that optimistic investors tend to overreact and, in the end, receive less of what they expected. Furthermore, another mechanism arises: conservatism, which "states that individuals are slow to change their beliefs in the face of new evidence" (Barberis et al. 1998). Then, investors, divided into optimistic and pessimistic traders, behave differently according to the weight they designate to

a particular announcement, and are unlikely to change their mind, even though a strong proof is supplied. This wrong assessment leads to persistent mispricing and a deterioration of the final wealth.

Baker and Wurgler (Baker and Wurgler 2006) argue that the issue of mispricing derives from an "uninformed" sentimental demand shock. According to behavioural finance, there is a strong debate on market efficiency, since the allocation of capital could be prone to encounter several risks (for example, fundamental and noise trader risk) during the investment and implies costs due to mispricing (Barberis and Thaler 2003).

## 2.2. Empirical Investigation

Various authors have contributed to influence the scientific field with a great number of papers regarding the investor sentiment and its effects. Hereafter, we provide a brief summary of the ones we consider the most worthy and appropriate previous studies on this topic.

Fisher and Statman (Fisher and Statman 2000) investigate three different groups of investors: individuals, newsletter writers, and Wall Street strategists. While the first two are almost perfectly correlated, there is no correlation of them with the last group. The study reveals that the future S&P500 returns have a negative and statistically significant relationship with individual investors and strategists of Wall Street.

Additionally, Brown and Cliff (Brown and Cliff 2005) prove that sentiment is negatively related to future returns. Then, if the investor sentiment is high (low), it will imply lower (higher) stock returns in the future. Smaller companies tend to be less affected by sentiment, while large firms even in long horizon are more influenced, with a consequently higher level of predictability power.

Baker and Wurgler (Baker and Wurgler 2006) explore the effect of the investor sentiment on cross section of stock returns. The results suggest that the sentiment is inversely proportional to stock returns—small, young, extreme growth, unprofitable, distressed, high volatility, and non-dividend-paying stocks. Another salient conclusion is that firm characteristics, that theoretically should not exercise any unconditional predictive power, show instead conditional patterns (for example, the U shape) as the sentiment is conditioned. This outcome can be explained as a compensation for the systematic risks, where some countermeasures, as the orthogonalisation of the investor sentiment index to macroeconomic circumstances, demonstrate inconsistency with this interpretation.

Baker and Wurgler (Baker and Wurgler 2007) examine in depth, theoretically and empirically, the investor sentiment, looking for an optimal way to measure it and to discern and quantify the consequences of it. They confirm that sentiment influences the cost of capital, with effects on the allocation of investments.

Lemmon and Portniaguina (Lemmon and Portniaguina 2006) investigate the time-series relationship between investor sentiment and stock returns using consumer confidence as a measure of investor optimism. Lemmon and Portniaguina (Lemmon and Portniaguina 2006) distinguish from a rational and an irrational part, the letter proxy by regression residuals. They find that a negative relationship between the sentiment and the stock market returns exists, even if a mispricing seems to be eventually corrected by noise traders.

From an international point of view, Schmeling (Schmeling 2009) researches if consumer confidence could have an impact on the expected stock returns in 18 industrialised countries. As before, Schmeling (Schmeling 2009) shows that sentiment has a negative relationship with forecasts of aggregate stock market returns. In addition, he provides a cultural explanation of why some countries have higher sentiment; indeed, most of them are more prone to overreact and to have a herding behaviour.

On the other hand, Verma and Soydemir (Verma and Soydemir 2006) point out that rational and irrational factors are both constituent parts of the investor sentiment, individual, as well as institutional. Furthermore, they brought to light a significant phenomenon: the contagion effect. The exploration consists of searching for an influence of one country's sentiment upon the assets of other markets. Their research evidences that the U.S. investor sentiment affects Mexico and Brazil, at an institutional stage, and U.K. at both the institutional and individual level.

Verma, Baklaci, and Soydemir (Verma et al. 2008) consider the impact of arbitrageurs and noise traders' sentiment on both the Dow Jones Industrial Average and the S&P500 returns. They find that irrational investor sentiment has a stronger effect on stock returns than rational one, justifying it with the speed of processing information about economic fundamentals.

Chung, Hung, and Yeh (Chung et al. 2012) also inspect investor sentiment in the business cycles and report that the predictability of the sentiment is meaningful only during the expansion, while in periods of recession there is no significance. Therefore, the investor sentiment results to be regime-dependent.

Huang et al. (Huang et al. 2014) propose a new investor sentiment, denoted as *aligned*, which outperforms the others, in terms of fitting, reducing incredibly the noise component, and predictability, with good results even in the out-of-sample forecasting method. Widely basing on the previous predictor of Baker and Wurgler (Baker and Wurgler 2006), they compare the results between the Baker and Wurgler (BW) sentiment and the aligned sentiment partially least square (SPLS).

Finally, Fernandes, Gonçalves, and Vieira (Fernandes et al. 2013) provide an examination of the "small" Portuguese stock market. Starting from the same hypothesis of the majority of the essays cited before, they investigate whether there exists predictability not only of aggregate stock returns, but also at industrial indices levels for Portugal, over the period 1997–2009. Using the residuals of the Economic Sentiment Indicator (ESI) for Europe and applying the principal component analysis technique to obtain macroeconomic factors, they document that sentiment shows a negative relation to returns. In addition, they inspect for the presence of a contagious effect of the U.S. investor sentiment on the local market.

## 3. Methodology

### 3.1. Indices and Models

Many different indicators have been proposed as investor sentiment index. Additionally, there are several different measurement mechanisms to build it. They can be divided mostly into two macro-categories: direct and indirect measures. Direct measures are all the indices, where the data are obtained through surveys conducted to consumers, investors or other agents, who explicitly give a response and their sentiment towards some specific questions and issues. The indirect measure is, instead, a financial or pure mathematical index used as a proxy to define the new sentiment indicator.

In the surveys, investors usually divide into bull, neutral, or bear. Alternatively, they are asked to express an opinion through numbers indicating high or low expectations. Some examples are the American Association of Individual Investors (AAII), which officially conducts and publishes surveys on investors; the Conference Board Consumer Confidence Index, which elaborates the surveys on individuals' expectations about issues in macroeconomics; and others that can deal with businesses or industrial sectors.

The literature provides many example of indirect measurements that can be assumed as sentiment indices. The more applied are: the IPOs, the number and average of first-day returns on Initial Public Offerings; NYSE turnover, measuring trading volume; CEFD, closed-end fund discount, since it seems to be inversely correlated to sentiment; dividend premium, which is the difference between average market-to-book ratios of payers and non-payers. All these proxies are considered as subject to sentiment, even though with probably different timing. Consequently, Baker and Wurgler, and Huang et al. (Baker and Wurgler 2006; Huang et al. 2014) combine more of these proxies to create one unique index.

Huang et al. (Huang et al. 2014) and before Baker and Wurgler (Baker and Wurgler 2006, 2007) study how the investor sentiment works and which factors are its constituents. Both indices are constructed from the same set of variables. Both the BW investor sentiment, created by Baker and Wurgler (Baker and Wurgler 2006, 2007), and the aligned one (here-hence denominated as SPLS), created by (Huang et al. 2014), are obtained from the following six individual sentiment proxies:

-      Close-end fund discount rate (CEFD);
-      Share turnover (TURN);
-      Number of IPOs (NIPO);
-      First-day returns of IPOs (RIPO);
-      Dividend premium (PDND); and
-      Equity share in new issues (EQTI).

In constructing the sentiment index, Huang et al. (Huang et al. 2014) and Baker and Wurgler (Baker and Wurgler 2006) use equal structure and the same choice of proxies (see above). The reference equation to create investor sentiment is written as follows:

$$Sent_t = CEFD_t\ \beta_1 + TURN_t\ \beta_2 + NIPO_t\ \beta_3 + RIPO_t\ \beta_4 + PDND_t\ \beta_5 + EQTI_t\ \beta_6 \tag{1}$$

Baker and Wurgler (Baker and Wurgler 2006) apply a first principal component, Huang et al. (Huang et al. 2014) prefer the partial least squares. According to (Huang et al. 2014), PC fails to produce significant forecasts because it can accumulate approximation errors coming from parts of the variations of the proxies. Hence, every one of the aforementioned proxy is moved on average with six months smoothing, standardised, and elaborated upon other regressions on industrial production, durable, and nondurable consumption, service consumption, employment, and a series of dummy variables in order to reduce the business cycle variation. In addition, the residuals coming from these regressions are used as proxy to be combined to build a new investor sentiment index. This procedure is the orthogonalisation to macro variables in order to compensate for systematic risk and to prevent high correlations, if the raw data are conditioned from macroeconomic factors.

Then, Huang et al. (Huang et al. 2014) apply a linear regression model where they regress sentiment indices at time t to predict returns at t + 1. We extend the regression in two directions. First, we include in the linear regression a set of control variables. Indeed, investor sentiment indices could proxy other information and we control for it. The resulting model is:

$$R_{t+1} = \alpha + \beta\ Sent_{t,k} + \delta X_t + \varepsilon_{t+1}, \varepsilon_{t+1} \sim i.i.d.\left(0, \sigma^2\right), k = 1, \dots, K \tag{2}$$

where $R_{t+1}$ is the excess market return at time t + 1, $Sent_{t,k}$ is the investor sentiment at time t, and k is one of the K alternative investor sentiment indices, $X_t$ is a set of predictors described in the next section. Second, we apply Bayesian inference. Barberis (Barberis 2000), Kandel and Stambaugh (Kandel and Stambaugh 1996), and Hodrick (Hodrick 1992) are among the first papers to advocate the use of Bayesian inference for investigating stock market predictability. Bayesian inference allows to set priors such as that the posterior distribution of the parameters of the predictive return regression can better learn from the data. This is particular useful when the sample size is small and priors help to reduce parameter uncertainty. Moreover, priors can be set to improve long-term asset allocation and to remove biases. Recently, Pettenuzzo, Timmermann, and Valkanov (Pettenuzzo et al. 2014) documented that economic constraints based on prior beliefs systematically reduce uncertainty about model parameters, reduce the risk of selecting a poor forecasting model, and improve both statistical and economic measures of out-of-sample forecast performance. We apply a normal-inverted gamma prior for our linear regression. We set prior mean values equal to OLS estimates and large prior variance values to keep the likelihood dominant on the prior. Degrees of freedom are set equal to 10% of the sample size. Our priors result in a closed form solution for parameter posteriors and predictive distributions. Precisely, the parameters β will follow a Student's *t* posterior distribution and the predictive density will also be *t*-Student distributed. See (Koop 2003) for exact values.[1] The estimation is run recursively.

---

[1]    We also investigate uniform flat priors. For the US example the results are almost identical; for the EU exercise we find large parameter uncertainties and lower forecast accuracy.

Up to the last observation posterior distributions and predictive densities are computed to predict the following value. In the next period, when new data are available, the process is repeated to obtain further predictions.

*3.2. Data*

The data span from January 1990–December 2014 (300 months) for the U.S. example, whereas the European example range from June 2001 through April 2017 (191 months). The European sample is unfortunately quite limited since the data are not available before the selected start point for all the components of the variables considered. As for the U.S. example, the length of the sample ends in 2014, because the data for the (Baker and Wurgler 2006, 2007) investor sentiment and the aligned investor sentiment calculated by (Huang et al. 2014) are available only until that year.

The dataset for the analysis in the U.S. market consists of the following variables:

- *Stock excess market returns of U.S. market*, SEMRUS: calculated from price of S&P500, including dividends and in excess of the risk free rate (3-month US treasury bill);
- *Continuous compounding of S&P500*, COMPOUND: calculated without dividends, in excess of risk free rate (10-year US treasury bill);
- *Investor sentiment index*, BW: calculated by (Baker and Wurgler 2006), through the PC method;
- *Orthogonalised investor sentiment index*, BWORT: calculated by (Baker and Wurgler 2006), the orthogonalisation is applied in order to reduce the systematic risk;
- *Aligned investor sentiment index*, SPLS: calculated by (Huang et al. 2014), through the PLS method;
- *Orthogonalised aligned investor sentiment index*, SPLSORT: calculated by (Huang et al. 2014), the orthogonalisation is applied for the same reasons as before;
- *Conference Board Consumer Confidence Index of US*, CB_CONS: calculated through surveys on expectations about business conditions, employment and income, from consumers over a six-month horizon;
- *CBOE's Volatility of S&P500*, VIX: annualised standard deviation, also known as uncertainty index, it is calculated from near expectations (one-month horizon) about stock market volatility.

Therefore, our sample includes four indirect measures and two direct measures of sentiment. The indirect measures of sentiment are downloaded from the Guofu Zhou website.

On the other hand, the dataset for the European consists of the following variables:

- *Stock excess market returns of EU market*, SEMREU: calculated from price of Euro Stoxx 50, including dividends and in excess of the risk free rate (3-month Euribor);
- *Continuous compounding of Euro Stoxx 50*, COMPOUND: calculated without dividends, in excess of risk free rate (10-year German government bond);
- *Economic Sentiment Indicator of European countries*, ESI_EU: published monthly by the European Commission, it consists of five sectoral confidence indicators (based on results from business surveys), which are: industry (40%), services (30%), consumers (20%), construction (5%) and retail trade (5%);
- *Economic Sentiment Indicator of Eurozone*, ESI_EUZONE: composite calculated only for the Eurozone countries;
- *Consumer Confidence Indicator of Europe*, CONSCONF: calculated from surveys on the financial situation of households, the general economic situation, unemployment expectations and savings, over one year horizon;
- *Industrial Confidence Indicator of Europe*, INDUCONF: calculated from surveys on production expectations, order books and stocks of finished products;
- *Economic Sentiment Indicator of Germany*, ZEW_DEU: calculated from surveys on expectations about macroeconomic development, financial and industrial profit situation over the following six months;

- *Ifo Business Climate Index*, IFO: dealing with the assessments of business situation and future expectations, it is calculated from surveys on different sectors from enterprises, such as manufacturing, construction, wholesaling and retailing, over a six-month horizon.

Therefore, the European example includes only direct measures of sentiment.

Our list of control for the U.S. stock market includes the 15 economic variables which are popular stock return predictors and are directly linked to economic fundamentals and risk aversion. We use the updated data From (Welch and Goyal 2008). Most of the predictors fall into four broad categories, namely: (i) valuation ratios capturing some measure of 'fundamentals' to market value such as the dividend yield, the earnings–price ratio, the 10-year earnings–price ratio or the book-to-market ratio; (ii) measures of bond yields capturing level effects (the three-month T-bill rate and the yield on long-term government bonds), slope effects (the term spread) and default risk effects (the default yield spread, defined as the yield spread between BAA and AAA rated corporate bonds, and the default return spread, defined as the difference between the yield on long-term corporate and government bonds); (iii) estimates of equity risk, such as the long-term return and stock variance (a volatility estimate based on daily squared returns); (iv) three corporate finance variables, namely the dividend payout ratio (the log of the dividend–earnings ratio), net equity expansion (the ratio of 12-month net issues by NYSE-listed stocks over the year-end market capitalization) and the percentage of equity issuance (the ratio of equity issuing activity as a fraction of total issuing activity). Finally, we consider a macroeconomic variable, inflation, defined as the rate of change in the consumer price index, and the net payout measure, which is computed as the ratio between dividends and net equity repurchases (repurchases minus issuances) over the last 12 months and the current stock price. As in (Welch and Goyal 2008), lag inflation is lagged an extra month to account for the delay in CPI releases.

For the European exercise, we could not collect all the 15 predictors and have eight variables: the dividend yield, the earnings–price ratio, the book-to-market ratio, the short-term interest rate, the long-term yield, the term spread, the default risk, the default return spread (where 10-years German bund rates are used as the government rate), stock variance (European VIX, VSTOXX), and inflation.

*3.3. Empirical Results*

3.3.1. The U.S. Market

The dependent variable is the excess market return, continuously compounded log return on the S&P 500 index (including dividends), minus the risk-free rate. The risk free rate is represented by the three-month U.S. Treasury bill.

Figure 1 shows four of the sentiment indices used for the U.S. market. Both the BW index and the SPLS have a similar pattern, since they are constructed starting from the same six variables, even though using different methods (PC and PLS, respectively). For this reason, the sentiment indices cannot be applied all together, but regress in separate equations.

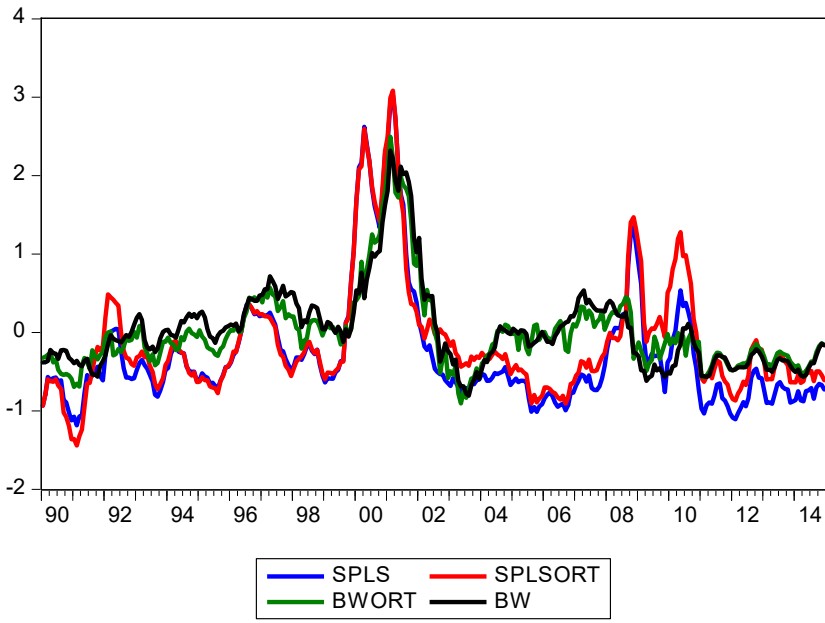

**Figure 1.** Plot of the sentiment indices group for the entire range of 1990–2014.

Figure 2 reports two sentiment indices, SPLS, and BWORT, and stock market returns. The figure documents that the latter variable is much more volatile than the sentiment, with great positive and negative peaks in short periods. As discussed in (Baker and Wurgler 2006, 2007), first, orthogonalisation applied to sentiment indices reduces the systematic risk. Second, the sentiment changes are more difficult to be detected, and its volatility expressed only in periods of high speculation.

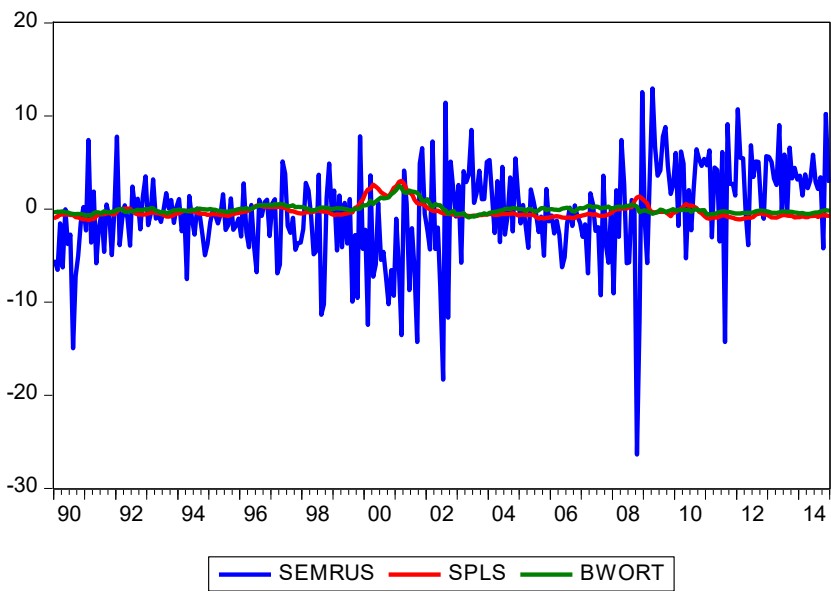

**Figure 2.** Plot of the sentiment indices, SPLS and BWORT, and the stock market returns, SEMRUS, for the entire range of 1990–2014.

Table 1 reports the results of the U.S. regression. The first four variables refer to indirect sentiment indices; the last two to (direct) consumer or market indices. All coefficients have negative posterior means, almost all the posterior mass has negative mass as the Bayesian *t*-statistics confirms and the posterior distribution assigns probability to positive numbers lower than 1%. The coefficient and the forecasts evaluation are consistent with the literature, proving that there exists a negative relationship between stock market returns and investor sentiment, supported by (Baker and Wurgler 2006, 2007;

Huang et al. 2014). Economically, one-percentage change in the independent variable is associated with an average decrease of −2.20 (for the BW) in the excess market return.

**Table 1.** Set of regressions run on the U.S. market.

| Variable | Post Mean ß | Bayesian *t*-Stat | Positive Post. Distr. | MSPE Ratio |
|----------|-------------|-------------------|-----------------------|------------|
| SPLS | −1.079 | −1.965 | 0.050 | 0.933 ** |
| BW | −2.200 | −3.149 | 0.002 | 0.926 ** |
| SPLSORT | −1.041 | −2.068 | 0.040 | 0.940 ** |
| BWORT | −2.350 | −3.318 | 0.001 | 0.939 ** |
| CB_CONS | −0.046 | −1.685 | 0.093 | 0.938 ** |
| VIX | −0.259 | −5.150 | 0.000 | 0.954 ** |

Note: This table reports the posterior mean of the sentiment indices used in the various regressions with US data; the Bayesian *t*-statistics, computed as the ratio of the posterior mean and the posterior standard deviation of the parameter; the probability of the positive posterior distribution. The last column gives the out-of-sample mean square error (MSPE) relative to the MSPE of the random walk benchmark. A MSPE ratio lower than 1 means that the alternative model based on the sentiment index outperforms the random walk benchmark. We measure statistical significance relative to the benchmark model using the (Diebold and Mariano 1995) *t*-tests for equality of the average loss. Asterisks indicate significance at * 10% and ** 5% levels. All results are based on the whole forecast evaluation period January 2008–December 2014.

As next step, we produce one-month forecasts from January 2008 to December 2014 using an expanding window approach. We compute mean square prediction errors (MSPE) by comparing each (point) forecast to the realization. We also compute forecasts using the standard benchmark model used for studying return predictability, the random walk in mean. We report MSPE ratios by dividing the MSPE of each of our models based on one of the sentiment indices by the MSPE of the benchmark. A MSPE ratio lower than 1 means that the alternative model based on the sentiment index outperforms the random walk benchmark. A MSPE ratio larger than 1 means that the benchmark is more accurate. We also test the difference of the MSPE based on the alternative model and the one based on the benchmark model using the (Diebold and Mariano 1995) test; see (West 1996) for theoretical foundations.

Among the sentiment indices, we find that BW provides the most accurate predictions of stock market returns with a reduction on MSPE relative to the benchmark of almost 7.5%. The difference is statistically significant. This result contrasts with (Huang et al. 2014), who found the SPLS being more accurate in the out-of-sample analysis. The SPLS is still statistically superior to the benchmark, but adding the control variables in (Welch and Goyal 2008) reduces marginally its economic gains. When testing the difference in MSPE of the models based on BW and SPLS indices, the null of equal predictability is not rejected.

When comparing to the other indices, we find that all models statistically outperform the benchmark model and the VIX variable gives the lower gains. Interesting, an index like the CB_CONS, which is made up of opinions and should be more inclined to errors, seems to be more appropriate (economically, but not statistically) to represent the investor sentiment, providing a larger predictability power than a financial variable as the VIX. We notice that the VIX and the stock variance in (Welch and Goyal 2008) dataset are highly correlated and this can create some imprecision on estimation. Finally we run two further set of regression models. The first one is a model based only on the (Welch and Goyal 2008) regressors and exclude the sentiment indices. The model gives a 3% reduction on MSPE relative to the benchmark, confirming the predictability power of the sentiment indices which all results on lower ratios. The second set of models removes control variables and apply the six sentiment indices individually in each regression. MSPE gains reduces, but BW and CB_CONS still provide a statistical significant reduction up to 5%, providing further evidence of their predictability power.

### 3.3.2. The European Market

In this section, we deal with the analysis and interpretation of the EU example. The excess stock market returns of the Euro Stoxx 50 is predicted through different European sentiment indices and a set of control variables.

Figure 3 plots the sentiment group on the entire sample, formed by the two Economic Sentiment Indicators, one for all Europe and one for the Eurozone, and the Consumer and Industrial Confidence Index. For making visible the trend of the series, the mean value is subtracted to the variables ESI_EU and ESI_EUZONE, levelling them to the other two indices. The two sentiment indices cannot be used all together because of multicollinearity issue since INDUCONF and CONSCONF are two of the five component sectors of the ESI.

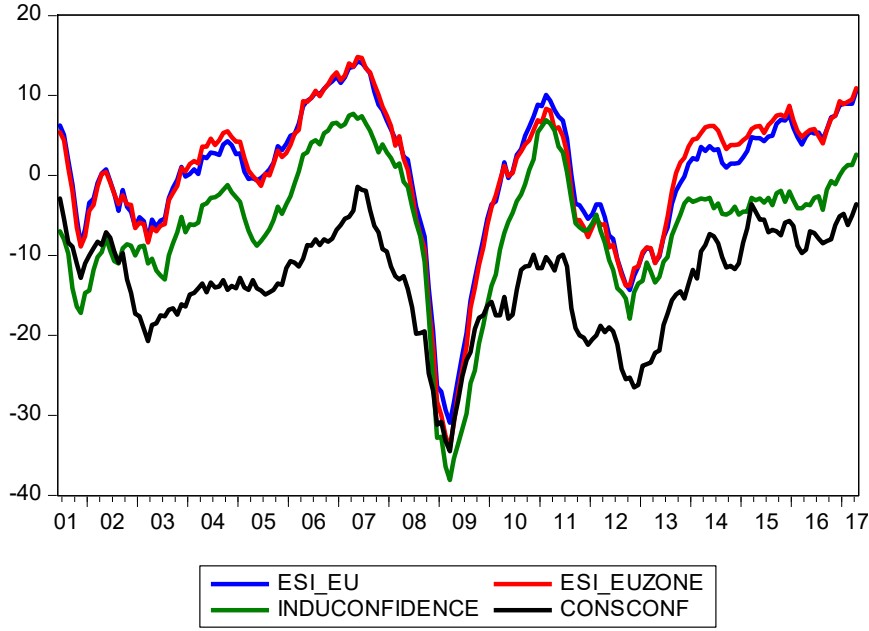

**Figure 3.** Plot of the sentiment indices group for the entire range 2001–2017.

Figure 4 show the volatile pattern of Euro Stoxx 50 compared to Consumer and Industrial Indices. At the end of the 2008 the negative peak in sentiment indices due to the financial crises is evident. On the contrary, on the same period, in particular the following months, stock returns recorded an increasing evolution with positive peaks.

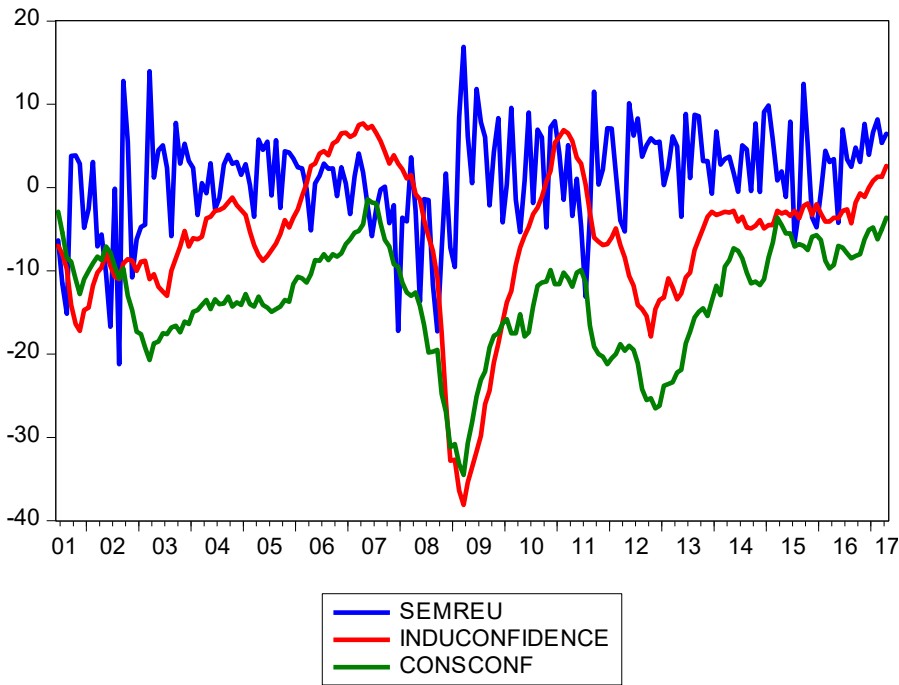

**Figure 4.** Plot of the economic indices, CONSCONF and INDUCONF, and the stock market returns, SEMREU, for the entire range 2001–2017.

Table 2 shows regression results. In this example, the Economic Sentiment Indicator and two specific components of it substitute the BW and SPLS indices. The economic indicators we choose are German industrial confidence index, ZEW_DEU and IFO indicators. The choice of these two German indices comes from different reasons. First, Germany is considered as a leading country in Europe, with a stronger economic and political stability. Second, Germany is an industrial and financial centre, with contacts to many European regions. Finally, the surveys reflect optimistic and pessimistic share for the future expected economic development not only in Germany, but also in France, Italy, and other relevant countries.

**Table 2.** Set of regressions on the EU market.

| Variable | Post Mean ß | Bayesian *t*-Stat | Positive Post. Distr. | MSPE Ratio |
|:---:|:---:|:---:|:---:|:---:|
| ESI_EU | −0.040 | −0.410 | 0.663 | 1.000 |
| ESI_EUZONE | −0.040 | −0.332 | 0.740 | 0.999 |
| CONSCONF | −0.490 | −2.690 | 0.006 | 1.000 |
| INDUCONF | 0.012 | 0.143 | 0.894 | 1.002 |
| ZEW_DEU | 0.013 | 0.723 | 0.463 | 1.036 |
| IFO | 0.052 | 0.503 | 0.584 | 1.030 |

*Note*: This table reports the posterior mean of the sentiment indices used in the various regressions with European data; the Bayesian *t*-statistics, computed as the ratio of the posterior mean and the posterior standard deviation of the parameter; the probability of the positive posterior distribution. The last column gives the out-of-sample mean square error (MSPE) relative to the MSPE of the random walk benchmark. A MSPE ratio lower than 1 means that the alternative model based on the sentiment index outperforms the random walk benchmark. We measure statistical significance relative to the benchmark model using the (Diebold and Mariano 1995) *t*-tests for equality of the average loss. Asterisks indicate significance at * 10% and ** 5% levels. All results are based on the whole forecast evaluation period January 2008–April 2017.

Except for the Consumer Confidence Index, posterior probabilities of other variables assign large probabilities to positive numbers. Therefore, apart from CONSCONF, there is no strong evidence on the role of sentiment indices to drive the EU stock market. From the economic point of view, this can be justified by the fact that Europe has not a strong financial impact comparable to the volumes of the U.S., which has been historically the leader of the worldwide markets. Fernandes, Gonçalves, and

Vieira (Fernandes et al. 2013) concluded that the Portuguese market has tendency to be affected by the sentiment, because of the high level of collectivism in the country. The herding is counterbalanced by the presence of institutional investors, which are considered as rational. This statement could lead to think that it is likely to notice a majority of rational investors in the EU market, because of institutional level, than noise traders. The forecasting exercise over the sample 2008–2017 confirms evidence and all models perform similarly in terms of MSPE. None models statically outperform the random walk benchmark.

### 3.3.3. Spillover Effect

Our forecasting sample deals with the period during and after the financial crisis, which had a global effect. Therefore, we investigate the possibility that the markets are not independent, where booms and recessions spread around different geographic regions.

Table 3 shows the results of predicting the European stock returns with U.S. sentiment indices. In these regressions, we exclude the set of control variables and just focus on the spillover effects. The output demonstrates a spillover effect for almost all the U.S. sentiment indices to European markets. Only the VIX does not support this evidence. The survey indicator CB_CONS is the only with a positive coefficient. BW produces the lowest MSPE, but it is not statistically significant relative to the benchmark model. One explanation for the result is that European investors misinterpret U.S. sentiment fluctuations, also due to the large capitalization of the U.S. market, confusing them with fundamental news and reacting to them in their European portfolio.

**Table 3.** Set of estimations run using the U.S. sentiment in order to predict the EU stock returns.

| Variable | Post Mean ß | Bayesian *t*-Stat | Positive Post. Distr. | MSPE Ratio |
|---|---|---|---|---|
| SPLS | −3.880 | −3.061 | 0.003 | 1.095 |
| BW | −3.195 | −3.055 | 0.003 | 0.990 |
| SPLSORT | −4.964 | −3.588 | 0.001 | 1.064 |
| BWORT | −3.283 | −2.902 | 0.005 | 1.018 |
| CB_CONS | 0.097 | 2.224 | 0.029 | 1.037 |
| VIX | −0.161 | −1.566 | 0.121 | 1.037 |

*Note*: This table reports the posterior mean of the sentiment indices used in the various regressions; the Bayesian *t*-statistics, computed as the ratio of the posterior mean and the posterior standard deviation of the parameter; the probability of the positive posterior distribution. The last column gives the out-of-sample mean square error (MSPE) relative to the MSPE of the random walk benchmark. A MSPE ratio lower than 1 means that the alternative model based on the sentiment index outperforms the random walk benchmark. We measure statistical significance relative to the benchmark model using the (Diebold and Mariano 1995) *t*-tests for equality of the average loss. Asterisks indicate significance at * 10% and ** 5% levels. All results are based on the whole forecast evaluation period January 2008–April 2017.

Table 4 reports the estimations of the U.S. stock returns through the European sentiment indices. As shown, all the variables have almost all the posterior mass in the negative support. Economically, it seems that the Economic Sentiment Indicator, elaborated by the European Commission, has a stronger link with American investors, enabling predictions on stock returns, than with the EU market. The model based on it outperforms the random walk benchmark at a 5% significance level. We notice that gains are, however, smaller than using U.S. sentiment indices. Zew and Ifo indices were not inserted in this table because of the unavailability of data for the entire range.

**Table 4.** Set of estimations run using the EU sentiment indices in order to predict the U.S. market.

| Variable | Post Mean ß | Bayesian *t*-Stat | Positive Post. Distr. | MSPE Ratio |
|----------|-------------|-------------------|-----------------------|------------|
| ESI_EU | −0.120 | −3.997 | 0.000 | 1.021 |
| ESI_EUZONE | −0.127 | −4.387 | 0.000 | 0.997 |
| CONSCONF | −0.191 | −4.862 | 0.000 | 0.976 * |
| INDUCONF | −0.117 | −3.550 | 0.001 | 1.060 |

*Note*: This table reports the posterior mean of the sentiment indices used in the various regressions; the Bayesian *t*-statistics, computed as the ratio of the posterior mean and the posterior standard deviation of the parameter; the probability of the positive posterior distribution. The last column gives the out-of-sample mean square error (MSPE) relative to the MSPE of the random walk benchmark. A MSPE ratio lower than 1 means that the alternative model based on the sentiment index outperforms the random walk benchmark. We measure statistical significance relative to the benchmark model using the (Diebold and Mariano 1995) *t*-tests for equality of the average loss. Asterisks indicate significance at * 10% and ** 5% level. All results are based on the whole forecast evaluation period January 2008 to December 2014.

To sum up, Tables 3 and 4 document a link between financial markets and the two markets are not independent, but interdependent.

## 4. Conclusions

This paper applies the sentiment index in regression models to predict US and European stock market returns. Many measurements are experimented, from direct sentiment indices, like surveys, to indirect measures of investor sentiment, such as the ones calculated by (Huang et al. 2014; Baker and Wurgler 2006, 2007). Differently than the previous literature, we control for a set of variables often used in return predictability and apply Bayesian inference to reduce parameter uncertainty due to the short sample, in particular for the European example.

As regards the American example, the results showed that globally sentiment has a negative impact on the stock market returns and BW resulted to have the highest predictive power. With respect to the European market, evidence shows weak findings and no relationship is found. Finally, the results show the presence of spillover effect between the two markets. Therefore, it can be concluded that U.S. and EU are two interdependent markets. In the end, this idea can justify the weak outputs on the European markets. From an economic standpoint, affected from globalisation and quick communication, Europe could be more prone to follow the influence of the American sentiment, because of the stronger economy.

Unfortunately, due to unavailability of data, the analysis is conducted on a limited range. The short period is a limitation on estimating the best model, since there could be omitted factors influencing the estimation. The use of Bayesian priors limits somewhat such effects. However, in future works it could be interesting to explore the difference between the rational and irrational factors of the sentiment, deepening the irrational analysis (i.e., the residual part).

**Author Contributions:** The work was equally divided between the two coauthors. The origin and development of the paper was a joint initiative. C.L.C. focused on collection of data and econometric analysis; F.R. worked on writing results and the working paper.

**Funding:** This work was supported by the Open Access Publishing Fund of the Free University of Bozen-Bolzano.

**Acknowledgments:** We thank the editor, two anonymous referees and Stefano Grassi for helpful comments and suggestions to improve this work. Chiara Limongi Concetto thanks the Italian Banking, Insurance, and Finance Federation to award the RIFET (Rome Investment Forum Empowers Talents) 2017 award.

**Conflicts of Interest:** The authors declare no conflict of interest.

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
