# Peer review of "Optimism in Financial Markets: Stock Market Returns and Investor Sentiments"

_jrfm, doi:10.3390/jrfm12020085_

Round 1

Reviewer 1 Report

Referee Report: “Optimism in Financial Markets: Stock Market Returns and Investor Sentiments”

Summary: Behavioural finance factors tend to be very important concepts which can drive fluctuations in stock markets. While this avenue has been amply tested in American data, this paper extends the potential relationship between the US and European sentiments. This paper takes stock that the economic sentiments tend to help to predict one step ahead returns. Optimism is captured using a range of methods, predominantly following Baker and Wurgler and the Sentiment Partially Least Squares of Huang, Jiang, Tu and Zhou (2014). Other confidence indices such as the VIX was also considered. For the European case, confidence indices measures were retrieved from European markets. Predictability of returns was also investigated using a battery of techniques with an application of Bayesian methods and other measures to judge the statistical significance of the predictability.

Comments : This is a very interesting topic to look at and there seems to be a decent amount of work undertaken for this paper. The amount of effort which led to the write up of these results are appreciable. However, I fear that this paper is seriously lacking in various respects, which I elaborate upon, with the view that the authors can develop their paper better.

(1)   Many parts of this work were poorly written and I am a bit at a loss understanding what results are being discussed upon, and which series is being commented on. This is further complicated by some wrong sentence structures. I provide some examples here, for the benefit of this paper

Pg 1, second paragraph: “ It is well known that investors are driven by fluctuations in sentiment, and before making an investment, they tend to behave differently”

Pg 8, first paragraph: “Bullish has a positive relationship due to the nature of the index” [Perhaps worth giving each of the variable on pg 6 a correct variable description.

2nd paragraph: Baker et al. (1992 ) happen to be misquoted. Obviously, the original authors made a mistake here with the word “on” which should have been “and”, but one should expect the current authors of this paper to pick this up !

These instances are numerous in this paper – The authors may want to spend considerable time proofreading the whole paper.

Presentation issues:

Moreover there are other issues with respect to the graphs and what they represent – what is ESI1 and ESI2.

 Explanation of the results – Readers may appreciate more structure here.

(2)   The motivation of this paper should be put in better focus. The first paragraph should be written to clearly express why optimism is different from fundamentals – for at the end of the day optimism could be nested within fundamentals simply as a factor moving demand and supply. Based on the literature, I am not sure what the authors tried to do with explaining what arbitrage means and noise traders are, within the context of this paper. Perhaps they could spare the space for explaining what the Bayesian approach is being used.

(3)   With respect to equation (1), it would be great to see the coefficients, purely out of curiosity as you would now have the financial crisis sample in which Baker and Wurgler did not have.

(4)   Moreover, the authors should be more explicit while delving into the sentiment extraction.

(5)   With respect to the comparison of table 1 and 3, it seems that the parameters tend to be lower on average for the European case, what could be the reason(s) for this – is it purely a sample size effect. On the other hand, we find that that the benchmark of random walk outperforms the sentiment indices. The authors should hold those results accountable, and probe a little bit deeper. I am afraid that the story about spillover may need to be better motivated.

(6)   While I feel the predictability story is interesting, it does not add more to the literature, and in the wake of more robust predictability testers, the authors may want to focus on only one theme: the spillover to motivate the story.

I feel that the paper has some potential, but it needs to be carefully polished

Author Response

Reply to referee 1 comments. Revised paper is entitled:

Optimism in Financial Markets: Stock Market Returns and Investor Sentiments

by C. Limongi Concetto and F. Ravazzolo

We thank the referee for the thoughtful suggestions and comments. We have made changes to the paper to incorporate the comments provided and we believe that the paper has substantially improved.

Please find below a point-by-point response to your comments (in blue).

This is a very interesting topic to look at and there seems to be a decent amount of work undertaken for this paper. The amount of effort which led to the write up of these results are appreciable.

Thanks.

However, I fear that this paper is seriously lacking in various respects, which I elaborate upon, with the view that the authors can develop their paper better.

 (1)   Many parts of this work were poorly written and I am a bit at a loss understanding what results are being discussed upon, and which series is being commented on. This is further complicated by some wrong sentence structures. I provide some examples here, for the benefit of this paper

Pg 1, second paragraph: “ It is well known that investors are driven by fluctuations in sentiment, and before making an investment, they tend to behave differently”

2nd paragraph: Baker et al. (1992 ) happen to be misquoted. Obviously, the original authors made a mistake here with the word “on” which should have been “and”, but one should expect the current authors of this paper to pick this up!

Pg 8, first paragraph: “Bullish has a positive relationship due to the nature of the index” [Perhaps worth giving each of the variable on pg 6 a correct variable description.

These instances are numerous in this paper – The authors may want to spend considerable time proofreading the whole paper.

We thank the referee for the effort to indicate weak sentence. We have corrected all them and now believe that the exposition and language of the manuscript has substantially improved. The paper has also professionally proofreading. We also removed sentences related to the role of arbitrage, which is not crucial to our analysis, see point 2) in the report too. And we removed the BULLISH indicator. The survey gives three different options to respondents: pessimistic, neutral and optimistic. The AAII website indicates that the relevant information is the changes across the three indicators and your comment makes clear to us that peaking just one of them is too restrictive for our analysis, thanks.

Moreover there are other issues with respect to the graphs and what they represent – what is ESI1 and ESI2.

The referee is right and we compiled a version with old names. We have substitute them and the series in the list on pages 9 and 10, Tables and Figures have all the same labeling.

Explanation of the results – Readers may appreciate more structure here.

We added more analysis to the results. Following your comments below and referee 2 suggestions, the text reports numbers when using a model with all regressors excluding sentiment indices and models based on individual sentiment indices and no regressors, see pages 13-14. This confirms the predictability role of sentiment indices.

Second, we try to address the different between US and EU results and the spillover effect. We think one possible reason is the “majority of rational investors in the EU market, because of institutional level, than noise traders.” Adding to the smaller capitalization of the European market and therefore a general lower appetitive for European stock helps to mitigate irrational behaviors.  

(2)   The motivation of this paper should be put in better focus. The first paragraph should be written to clearly express why optimism is different from fundamentals – for at the end of the day optimism could be nested within fundamentals simply as a factor moving demand and supply. Based on the literature, I am not sure what the authors tried to do with explaining what arbitrage means and noise traders are, within the context of this paper. Perhaps they could spare the space for explaining what the Bayesian approach is being used.

We have added the definition of optimism in the first paragraph: “Optimism, also known as market sentiment, reveals the movements in the financial markets dictated by the psychological perception of determined operations or trades”. We believe the definition clarifies why optimism is different from economic fundamentals and optimism biases return expectations.

We have also removed the arbitrage story and reference to behavioral finance to focus on the issue of definition of sentiment.  

(3)   With respect to equation (1), it would be great to see the coefficients, purely out of curiosity as you would now have the financial crisis sample in which Baker and Wurgler did not have.

We computed coefficients in the sample 1990M1-2007M12 using OLS estimation and not regressors. The sample corresponds to our first in-sample period, it excludes most of the financial crisis and we removed prior information and regressors to be similar as much as we can to Baker and Wurgler. Coeffiencents are reported in the table below and are all significant at 1% level.

The role of sentiment indices is confirmed and evidence is qualitatively similar to the full sample analysis. Therefore, it is not only a financial crisis effect, but it seems a more persistent relantionship.  

SPLS

-1.247

BW

-1.978

SPLSORT

-1.122

BWORT

-2.028

CB_CONS

-0.043

BULLISH

16.220

VIX

-0.198

(4)   Moreover, the authors should be more explicit while delving into the sentiment extraction.

We have tried to emphasize more how indices are selected and computed. The new labelled section “3.1 Indices and Models” divides indices in direct and indirect measures and provides detailed description on how to compute indirect indices. The following section “3.2 Data” describes the indices used in our paper. Direct indices are all available in the web from standard libraries; indirect indices are download from Guofu Zhou website as we report in page 9.

(5)   With respect to the comparison of tables 1 and 3, it seems that the parameters tend to be lower on average for the European case, what could be the reason(s) for this – is it purely a sample size effect. On the other hand, we find that that the benchmark of random walk outperforms the sentiment indices. The authors should hold those results accountable, and probe a little bit deeper. I am afraid that the story about spillover may need to be better motivated.

Coefficients in Tables 1 and 3 are not exactly comparable for three main reasons. First, the sample is different as the referee indicates. Second, the dependent variable is different. Third, the set of controls is different. In the US example in Table 1 we include the 15 economic variables in the updated data from Welch and Goyal (2008); in the EU example in Table 3 we do not include control variables. We also do not include control variables in the regression for Table 4; whether we include 8 controls in the regression for Table 2. Our decision to exclude control variables for the spillover analysis in Tables 3 and 4 is motivated by narrowing the focus on the spillover. Indeed, when adding control variables in the US regression for Table 4 we find evidence of predictability, which is driven by the controls and not the EU sentiment indices.

(6)   While I feel the predictability story is interesting, it does not add more to the literature, and in the wake of more robust predictability testers, the authors may want to focus on only one theme: the spillover to motivate the story.

We think it is important to document that sentiment indices have predictability power for the US market, but not for the EU market. We motivate the result as a majority of rational, because mainly institutional, investors in the EU market. We think the finding is new in literature. However, US sentiment indices predict EU market. One possible explanation that we added in the text is that European investors misinterpret US sentiment fluctuations, also due to the large capitalization of the US market, confusing them with fundamental news and reacting to them in their European portfolio.  

Reviewer 2 Report

This is clearly an interesting paper that fits very well the scope of JRFM. The author expand previous research with respect to the predictive power of sentiment indices in US and Europe for stock market returns. They innovate by considering enlarged specifications for the predictive regressions and through the use of Bayesian methods

While there is a clear contribution, I also suggest that the author should try to improve the paper. Here are my comments:

The author should compare their findings with the findings based on the standard regressions, without controls. This is important, since it is one of the contributions.

What if the authors use classical inference, this could be added and it would reinforce the arguments of the authors.

Why would the results differ between US and Europe? Are there structural differences? What could an academic or investor learn from these results?

Author Response

Reply to referee 2 comments. Revised paper is entitled:

Optimism in Financial Markets: Stock Market Returns and Investor Sentiments

by C. Limongi Concetto and F. Ravazzolo

We thank the referee for the thoughtful suggestions and comments. We have made changes to the paper to incorporate the comments provided and we believe that the paper has substantially improved.

Please find below a point-by-point response to your comments (in blue).

This is clearly an interesting paper that fits very well the scope of JRFM. The author expand previous research with respect to the predictive power of sentiment indices in US and Europe for stock market returns. They innovate by considering enlarged specifications for the predictive regressions and through the use of Bayesian methods

Thank you.

The author should compare their findings with the findings based on the standard regressions, without controls. This is important, since it is one of the contributions.

Thanks for this remark. We agree and we have added these numbers at the end of section 3.3.1 with the following sentence:

“The second set of models removes control variables and apply the seven sentiment indices individually in each regression. MSPE gains reduces, but BW and CB_CONS still provide statistical significant reduction up to 5%, providing a further evidence of their predictability power.”

What if the authors use classical inference, this could be added and it would reinforce the arguments of the authors.

This is a very good suggestion and we computed numbers using a frequentist approach, see Tables below for the US and the EU respectively. By comparing them with Tables 1 and 2 in the paper, we conclude that Bayesian inference helps providing reductions around 1.5-2%, even if sentiment indices still provide lower MSPE figures than the benchmark for the US. At the end, we decided not to report these numbers in the paper for at least two reasons. First, we shall double tables, but evidence is similar with sentiment indices having predictability power for the US and not for the EU. Second, our samples are somewhat shorter for the lack of data on sentiment indices. But data for Welch and Goyal (2008) regressors go back to post-war II period. Bayesian reduces parameter uncertainty, but we speculate that if the sample was longer, results would be similar.   

US results  Variable

RMSE

SPLS

0.959

BW

0.941

SPLSORT

0.983

BWORT

0.944

CB_CONS

0.957

BULLISH

0.994

VIX

0.991

EU results

Variable

RMSE

ESI

1.015

ESI_EUZONE

1.014

CONSCONF

1.018

INDUCONF

1.016

ZEW_DEU

1.051

IFO

1.045

Why would the results differ between US and Europe? Are there structural differences? What could an academic or investor learn from these results?

We try to address this comment at the end of section 3.3.2 and believe that “a majority of rational investors in the EU market, because of institutional level, than noise traders.” Adding to the smaller capitalization of the European market and therefore a general lower appetite for European stock helps to mitigate irrational behaviors.  

Round 2

Reviewer 1 Report

This is a significantly improved paper. Most of my queries have been addressed. However, I still think that this paper is seriously lacking in terms of how it is written and presented. I would recommend the authors to have the paper professionally edited. 

Reviewer 2 Report

The authors have replied in both writing and by modifying the paper accordingly to all the issues raised in the first round of review. I suggest the acceptance of the paper.

This manuscript is a resubmission of an earlier submission. The following is a list of the peer review reports and author responses from that submission.

Round 1

Reviewer 1 Report

Please see my referee report.

Reviewer 2 Report

The paper extends the model tested by Huang et al., (2014) using  a Bayesian approach 

My comments

1. The contribution of the paper is not clear for the reader and should be improved.

2. There are many grammatical mistakes

3. The authors argue that " The first intent of this section is to replicate the model tested by Huang, Jiang, Tu, and Zhou (2014) on a more recent dataset"  

They consider data from 1990 to 2014, not so recent ... and they do not explain why this choice?

4. The use of the Bayesian inference is not well explained and detailed. Is there preliminary tests for the choice of the priors? How do the authors make sure that the algorithm converge?

      5. The results are very poor and no implication is made for      

        investors.

This article should be improved in order to be published